# Postulating the Unicity of the Macroscopic Physical World

**DOI:** 10.3390/e25121600

**Published:** 2023-11-29

**Authors:** Mathias Van Den Bossche, Philippe Grangier

**Affiliations:** 1Thales Alenia Space, 26, Avenue J.-F. Champollion, 31037 Toulouse, France; mathias.van-den-bossche@thalesaleniaspace.com; 2Laboratoire Charles Fabry, IOGS, CNRS, Université Paris Saclay, 91127 Palaiseau, France

**Keywords:** quantum physics, contextuality, operator algebra

## Abstract

We argue that a clear view of quantum mechanics is obtained by considering that the unicity of the macroscopic world is a fundamental postulate of physics, rather than an issue that must be mathematically justified or demonstrated. This postulate allows for a framework in which quantum mechanics can be constructed in a complete mathematically consistent way. This is made possible by using general operator algebras to extend the mathematical description of the physical world toward macroscopic systems. Such an approach goes beyond the usual type-I operator algebras used in standard textbook quantum mechanics. This avoids a major pitfall, which is the temptation to make the usual type-I formalism ’universal’. This may also provide a meta-framework for both classical and quantum physics, shedding new light on ancient conceptual antagonisms and clarifying the status of quantum objects. Beyond exploring remote corners of quantum physics, we expect these ideas to be helpful to better understand and develop quantum technologies.

## 1. Introduction

### 1.1. Our World Is Unique…


Obvious empirical evidence tells us that we live and die in a single world, which has a history before and after our individual existence. Alternative histories of the Universe are possible but remain counterfactual and a subject of fiction. In a similar vein, the past can be forgotten but not changed, and the future can be predicted and depends, to some extent, on our actions or the absence of actions. It is actually a mix of quasi-certainty (the sun will rise tomorrow) and inherent uncertainties (it is likely to rain tomorrow). And once something has happened, good or bad, there is no way back. This obvious empirical evidence has set the framework in which mankind has evolved, and it has essentially not changed, despite the fact that our abilities to record the past and predict the future have tremendously changed over the centuries and millennia. Actually, most of our current techniques for recording, sending, and processing—and also manipulating—information use microelectronics, which itself is based on quantum physics.

Here, we want to argue that considering the unicity [1] of the macroscopic world as a basic postulate for physics is not only possible but appears as a firm basis on which quantum mechanics (QM) can be built. So, our approach does not contradict QM as it is currently used, but it embeds the usual quantum formalism in a framework where it is still valid, but where some misleading extrapolations do not show up [2]. From the mathematical side, this extended framework is not really new and may be traced back to John von Neumann at the end of the 1930s [3,4]. This framework is also used in mathematical physics when addressing some aspects of statistical physics and quantum field theory [5]. However, it is conceptually and technically demanding and has been mostly ignored by physicists in standard QM, especially in elementary textbooks. But, as we will see, it may be extremely useful to establish the overall consistency of QM.

### 1.2. … But Is It Classical and/or Quantum?

Since the definition of the theoretical framework of quantum mechanics to describe the microscopic world was initiated by Heisenberg and Schrödinger and completed by Dirac and von Neumann, the tremendous reliability of its predictions combined with the exotic phenomena it unveils has raised much interest in the scientific community and beyond. The exotic aspect has led to interpretations that many consider far-fetched, and mainstream quantum physicists tend to stick to a ’don’t bother and calculate!’ approach. We will call ’textbook quantum mechanics’ (TBQM) the formalism on which the mainstream approach relies [6,7,8]. Despite its efficiency, this attitude may not be the one that will allow much future development, especially in the frame of, e.g., emerging quantum technologies, as well as quantum gravity.

The main problem that motivates non-mainstream interpretations lies in the non-deterministic aspect of quantum projective measurements [2,9]. This aspect, when related to quantum superpositions, leads to such wordings as ’the system is in two states at the same time’ in popular texts close to mainstream thoughts. More divergent schools consider that QM could, in fact, be deterministic at the cost of hidden mechanisms (Bohmian mechanics, pilot wave) or at the expense of a proliferation of replicas of the Universe every time a projective measurement happens (Everett’s multiverse interpretation). An overview of the interpretations is presented in [2], chap. 11, and see also [10] It is true that the contrast between the smooth, predictable, unitary evolution of quantum systems without measurement on the one hand, and the abrupt, discontinuous, dissipative, random effect of measurement on the other creates shock waves that leave no one indifferent.

A second problem lies in the significant difference between the laws that apply to microscopic vs. macroscopic systems, all the more because macroscopic systems are made up of a very large number of microscopic systems. This leads to the idea that macroscopic laws should emerge from the laws of quantum physics as a sort of averaging out of quantum effects. This is the essence of a form of ’reductionism’, and it is true that Ehrenfest’s theorem or the properties of coherent states [11] seem to show a path in this direction. Nevertheless, another view, coming mostly from Bohr and Heisenberg and adopted in TBQM, tells us that there is an irreducible cut between the microscopic and the macroscopic worlds. TBQM can be considered ’dualist’ when considered from this point of view.

On the reductionist side, Zeh [12] and Zurek [13] have elaborated on the concept of decoherence, which explains how, during the process of measurement, the microscopic degrees of freedom of the quantum system of interest become entangled with a huge number of external degrees of freedom (in the measurement device, in its environment). The global state cannot be realistically tracked, thus leading to a leak of information outside the system. This leak can, however, be computationally managed in the density operator of the system and its environment by performing a partial trace on the environmental degrees of freedom. This leads to a situation where this operator becomes diagonal for pairs of associated system and device states. Although this does not explain how a precise measurement result is selected from among the possible ones, at least it shows how the system and device can couple to produce a result.

A third problem of a different nature has to be mentioned at this point. Our civilisation is on the verge of developing quantum technologies, which rely on using all the specific aspects of quantum physics, and not only some of them, as in electronics and photonics, where a part of the weirdness is hidden by the large quantities of involved electrons or photons. These technologies will be developed by engineers who will be all the more efficient if they can develop an intuitive understanding of QM. This requires moving beyond the various ad hoc and often wobbly explanations that are given for its basic properties, often by awkwardly trying to connect them with a typical, classical view of the world. This means that clarifying ’what exists’ (ontology) and what happens regardless of the observer (objectivity) is not only a philosophical question but is also a condition for the progress of these technologies.

The present article supports and provides a perspective on the idea that the ’Contexts, System, Modalities’ approach (CSM) developed in previous papers by one of the authors and their coworkers [14,15,16,17,18,19,20,21] may provide such a clarification of the understanding of QM.

The purpose of the present article is firstly to summarise the main features of CSM (Section 2), which starts with a few empirical postulates, including the unicity of the physical macroscopic world and the contextuality of measurements, in order to build up the formalism of TBQM. In order to present an overview of CSM, we will not provide the details of the proofs here but refer to published papers. Some major steps in the CSM framework include showing that quantum theory must be probabilistic and that the description of macroscopic systems requires an *operator algebraic framework* that is broader than the one traditionally used in TBQM. This theoretical extension actually provides an understanding of the classical-quantum transition and allows for building a comprehensive overall framework to embed both realms. In Section 3, we discuss several implications of this approach on a more general level, i.e., the validity of considering mathematically infinite systems and hazardous predictions related to the so-called ‘universal unitarity’. We will also argue that the reductionist vs. dualist views of physics may not be so antagonistic after all.

## 2. Overview of the Construction of Non-Fully Unitary QM

### 2.1. Introduction and Motivation

The heart of the conceptual difficulties of the quantum mechanism is in the measurement process, which is framed in TBQM by the von Neumann projection postulate [6,7,8]. What is the problem with this postulate? It is not inherently wrong since it is at the core of the functioning of TBQM and has been observed correctly whenever relevant observations could be made. The main typical criticism is that it introduces a projection—a non-unitarity evolution during measurement—seemingly contradicting the fact that the system and measurement apparatus should globally evolve according to the Schrödinger equation, which predicts a unitary evolution.

There are many ways to deal with this unsatisfactory dichotomy, most of which are based on the decoherence theory [12,13], which essentially states that measurement devices are large systems coupled to an even larger environment, where interactions ultimately create an extremely large entangled system. Therefore, it is not possible to keep track of all degrees of freedom, and unitarity is broken by taking a partial trace over the degrees of freedom that become out of control. In addition to this loss of information, the structure of the interaction with the environment selects the measurement basis through environment-induced selection, that is, ‘einselection’ [13].

After some reasonable approximations, this leads to a classical probability distribution over the possible results in the einselected basis, but one more problem arises: only one result is observed rather than a probability distribution of them. So, what does ‘select the winner’ mean in an individual measurement event? This question is more tricky because it assumes that there should be such a selection process, which does not exist in basic QM. So, a simple way to avoid the problem is to state that the result fundamentally exists as a probability distribution and that there is no such thing as a winner selection process. Then, the unique result can simply be observed at the macroscopic level, and the probability distribution is actualised as usual [22].

### 2.2. The Proposed Construction

Clearly, this observation plus actualisation process is fully consistent with a postulate on the unicity of the physical macroscopic world, and thus of the macroscopic measurement result. The CSM framework [14,15,16,17,18,19,20,21] is built upon this idea, with the following postulates:**P${}_{0}$** 
**Unicity of the macroscopic world** —There is a unique macroscopic physical world in which a given measurement yields a single result.**D${}_{1}$** 
**Contexts, systems, and modalities**—Given a microscopic physical system, a *modality* is defined as the values of a complete set of physical quantities that can be measured in this system. This complete set of physical quantities is called a *context*, and a modality is attributed to a system **within** a context. Contexts are concretely defined by the settings of macroscopic measurement devices.**P${}_{1}$** 
**Predictability and extravalence**—Once a context is defined and the system is prepared in this context, modalities can ideally be predicted with certainty and measured repeatedly in this system. When changing the context, modalities change as a general rule, but some modalities belonging to different contexts may be connected with certainty; this property is called extracontextuality, and it defines an equivalence class between modalities, called extravalence [14,15].**P${}_{2}$** 
**Contextual quantisation**—For a given system and context, there exist at most *D* distinguishable modalities that are mutually exclusive: if one modality is realised in an experiment, yielding a result in the macroscopic world, the other ones are not realised. The value of *D*, called the dimension, is a characteristic property of the quantum system and is the same in all relevant contexts.**P${}_{3}$** 
**Changing context**—Given P${}_{1}$ and P${}_{2}$, the different contexts relative to a given quantum system are related to each other through continuous transformations (e.g., rotating a polarisation beamsplitter), which are associative, have a neutral element (no change), and an inverse. Therefore, the set of context transformations has the structure of a continuous group, which is generally non-commutative.

For the sake of clarity, let us note that within the usual QM formalism (which we have not yet discussed), a complete set of commuting observables (CSCO) corresponds to a context, and a state vector (or projector onto that vector) corresponds to an extravalence class of modalities but **not** to a particular modality since the specification of the context is missing in the state vector as soon as D≥3 [14,15].

The crucial postulate P${}_{2}$ (contextual quantisation) can be understood as the consequence of dealing with the smallest bits of reality, which, for this reason, only have a finite quantity of information to give regardless of the way they are interrogated [23]. This is also related to the existence of truly indiscernible microscopic objects: if there was an infinite amount of information that could be carried by individual quantum objects, they would ultimately differ in at least one of these pieces of information. But, both theoretical and empirical evidence tells us that QM does not work this way.

Then, the leading idea of CSM is to start with these physical postulates, which involve the quantisation of the properties of microscopic systems within macroscopic contexts, and to first show that a probabilistic description is required to avoid contradiction [17]. The basic idea is that the results from different contexts cannot be gathered together, as this would lead to more than *D* mutually exclusive modalities, contradicting P${}_{2}$. Therefore, the link between modalities in different contexts *must be probabilistic*: given a modality in a first context, only the probability of obtaining another modality in a different context can be predicted [16]. Additionally, it appears that this probabilistic aspect and its underlying indeterminism are key to maintaining relativistic causality when spatially extended contexts are considered. As a matter of fact, thanks to them, only randomness, i.e., non-information and entropy, can be transferred over space-like intervals. This is related to the discussion on predictive incompleteness [24] presented in [18] (see, e.g., Figure 1 in this reference).

It can also be shown that typical classical probabilities are not suitable to warrant that (i) there is a fixed number of mutually exclusive modalities in any context, and (ii) that the certainty of extravalent modalities can be transferred between contexts. In TBQM, this is most directly shown by the Kochen–Specker theorem [25,26]. A more suitable framework (worth trying!) is then to associate mutually orthogonal projectors with the events corresponding to mutually exclusive modalities [17,27].

**P${}_{4}$** 
**Projective probabilities**—In a given context C, each exclusive modality ${\left\{m_{i}\right\}}_{i=1}^{D}$ of a system is represented by a projector ${\hat{\Pi }}_{i}$ in a Hilbert space of dimension *D*, with all ${\hat{\Pi }}_{i}$’s being orthogonal.

Then, the above postulates can be used to justify the hypotheses of powerful mathematical theorems, which are, respectively, Uhlhorn’s theorem for connecting different contexts with unitary transformations [28,29] and Gleason’s theorem [30,31] for deriving Born’s rule [14]. The projectors are thus the basic mathematical tools related to probabilities, and from them, it is easy to construct standard observables:**D${}_{2}$** 
**Observables as operators**—From the orthogonal rays generated by the ${\hat{\Pi }}_{i}$s, Hermitian operators on a *D*-dimensional Hilbert space can be constructed by considering each ${\hat{\Pi }}_{i}$ as an eigenspace associated with $m_{i}$, the corresponding eigenvalue. If $m_{i}$ corresponds to a single observable quantity, this yields an operator $\hat{M}=\sum_{i=1}^{D}m_{i}{\hat{\Pi }}_{i}$. If $m_{i}$ is a tuple of several observable quantities, a tuple of operators can be constructed in a similar way.

One thus recovers the standard definition of observables in a complete set of commuting observables (CSCO) in relation to the spectral theorem. In this framework, traces over products of projectors and observables thus emerge naturally as a way of computing an experimental expectation value of the said observable.

Again, the intuitive idea behind the postulates is that performing more measurements in QM (by changing the context) cannot provide more details about the system because this would increase the number of mutually exclusive modalities, which contradicts P${}_{2}$. One might conclude that changing the context totally randomises all results and that nothing can be predicted, but this is not true either: some modalities may be related with certainty across different contexts. This is why extravalence is an essential feature of the construction —actually, extravalent modalities tie the other, non-extravalent modalities to a predictable probability distribution through Gleason’s theorem [14,17].

The final step in moving beyond the typical TBQM is to apply this formalism to a countable infinity of systems with infinite tensor products (ITP) of elementary Hilbert spaces to model the macroscopic limit. By doing so, one moves from the separable Hilbert spaces and type-I operators considered in TBQM towards non-separable Hilbert spaces and the type-II and type-III operators described by Murray and von Neuman in the 1930s and 1940s [4]. In this limit, unitary equivalence and unitarity overall are lost [5], and the predicted behaviour looks very much like the classical one [19,20,21,32,33] (note that the latter reference agrees with us on the mathematical side, though not on the physical side). Therefore, the overall mathematical description aligns with the initially postulated separation between microscopic systems and macroscopic contexts (Heisenberg cut), closing the loop of the construction (Figure 1).

### 2.3. Discussion

We note that in the above approach, there is no need to call for partial traces or loss of information since decoherence is initially incorporated by the postulates and eventually recovered from the (mathematically) infinite character of the context; this full loop is thus self-consistent. It is also quite possible to make type-I calculations, for instance, to calculate decoherence times in a given experiment; but it should be made explicit that they are approximations that are able to come very close to the actual non-unitary jump but are unable to manage it. On the other hand, the overall framework sets a clear separation between the microscopic (system) level and the macroscopic (context) level and makes sure that there is nothing like super-contexts or some variety of Wigner’s friend that would be able to turn an unbounded context back into a system [16]. Similarly, reasonings based on a universally extended unitary evolution do not align with our framework because unitarity itself is lost in the limit of an infinite extension.

### 2.4. The Crucial Role of Unitary Transformations

Given the above statements, it is important to provide more details on what unitary transformations are and are not according to the CSM approach. In standard QM, unitary transformations have a variety of different roles. A standard one is time evolution, which we discuss below. In relation to the previous sections, one may look at a related issue, that is, the role of unitary transformations in quantum measurements. This turns out to be quite important in the CSM framework since a modality must be associated with a certain and repeatable result in a given context. However, this is not the most frequent situation in practical QM. Actually, in most cases, a unitary transformation is inserted into the quantum measurement itself.

For instance, let us discuss how to perform a measurement that yields a certain and repeatable modality in the following situations: (i) a coherent state |α〉 of a harmonic oscillator (or quantised electromagnetic field mode), (ii) a Bell state for two spins, and (iii) an arbitrary state of a quantum register. Looking first at the coherent state |α〉, it is clear that neither photon counting nor coherent detection will do the job: the first yields a Poisson distribution of photon counts and the second yields an amplitude value, with some probability distributions depending on how the measurement is implemented. But, obtaining the required certainty can be easily guessed. Let us deterministically translate |α〉 by (−α), obtain the vacuum |0〉, count zero photons with certainty, and translate back by (+α) to the initial state. The modality criterion is thus respected, but it is clear that the irreversible part of the measurement (the photon counting) is inserted between two reversible unitary transformations, here, translations.

A similar situation appears when carrying out a measurement in the Bell state basis with the four entangled states {|++〉±|−−〉,|+−〉±|−+〉}. One then has to use a CNOT gate and then a Hadamard gate, measure the spins along *z* in the factorised basis {|++〉,|−−〉,|+−〉,|−+〉}, and go back to the Bell basis by using the reverse unitary transform. This can obviously be generalised to an arbitrary quantum state of a register with many qubits. If such a state is prepared from the unitary transform $\hat{U}$, then apply ${\hat{U}}^{†}$, check that the register is back to its initial state (all zeros, for instance), and return to the arbitrary quantum state by applying $\hat{U}$ again.

Obviously, these examples are highly idealised since they assume perfect unitary transforms and perfect quantum non-demolition (QND) measurements when the irreversible check is performed. They are, however, perfectly legitimate from a quantum point of view and correspond to how a perfect quantum computer should work. The current gate fidelities ensure that the successive application of $\hat{U}$ and ${\hat{U}}^{†}$ cannot efficiently revert the system to its initial state unless a very small number of qubits is used—performing this with a very large register is extremely challenging although not forbidden in principle.

In the logic of CSM, these examples make it clear that unitary transformations describe the deterministic evolution or manipulation of isolated quantum systems within classical contexts outside the measurement periods [19,20]. However, they do not apply to the Universe as a whole. The contexts themselves are classically described and do not correspond to mathematical entities that can be the object of unitary transformations. In contrast, within the algebraic framework discussed here, they correspond to separate sectors that are not connected by any operator constructed at the system level [21]. This again illustrates the overall consistency of CSM from the previous physical postulates to the mathematical formalism and back.

## 3. Higher-Level Implications

The scheme summarised above establishes the TBQM formalism through a few postulates and then closes the loop by showing how the ITP limit reestablishes the key assumptions. This construction calls for examination from a broader perspective, particularly in three aspects: (i) the acceptability of the infinite system limit, (ii) the key role of unitarity and where it cannot be applied, and (iii) a new perspective on the debate between those who consider that classical physics emerges from quantum theory (reductionism) and those who think that there is a fundamental distinction (dualism). Actually, both positions might be *equivalent*. We discuss these three aspects in this section.

### 3.1. Is Infinity Acceptable at All?

Using infinity in a physical theory legitimately raises relevant questions. These questions relate intuitively to whether our Universe is infinite or not, which we do not know. Moreover, in the specific case we are considering, something even more puzzling occurs. At the infinite subsystem number N→∞ limit, the ITP of *N* elementary Hilbert spaces $H={⊗}_{\alpha =1}^{N}H_{\alpha }$ appears to suddenly become non-separable, i.e., qualitatively different, which could lead to the idea that anything valid at N<∞ no longer holds in the limit. Yet, there are two points one can make to justify taking the limit seriously: one is mathematical and the other is more epistemological.

*On the mathematical side*, looking at the details of von Neumann’s breakdown theorem [20,21], things are much more subtle than just a function that would be discontinuous at the limit. As a matter of fact, two key properties of non-separable ITP Hilbert spaces are:**(1)** 
The breakdown into non-unitarily equivalent orthogonal sectors that correspond to an infinite number of changes in the elementary subsystem states.**(2)** 
The fact that sectors are not connected by operators in the ring $B^{\#}$ built as an extension to the full ITP of operators that act on elementary subsystem Hilbert spaces, their products, sums, and topological completions.

Quite importantly, it can be shown that these two properties build up gradually. More precisely, ref. [3] shows the following:**(1)** 
If $|\Psi \rangle :={⊗}_{\alpha =1}^{N}|\psi_{\alpha }\rangle$ and $|\Phi \rangle :={⊗}_{\alpha =1}^{N}|\phi_{\alpha }\rangle$ in H are not in the same sector when N→∞, for any ε>0, one can find a finite set J⊂[1,…,N] of *M* indices αs, all distinct, so as to build $|\Psi_{M}\rangle :={⊗}_{\alpha \in J}|\psi_{\alpha }\rangle$ and $|\Phi_{M}\rangle :={⊗}_{\alpha \in J}|\phi_{\alpha }\rangle$, such that $|\langle \Psi_{M}|\Phi_{M}\rangle |<\varepsilon$**(2)** 
Assume $\hat{A}$ is a bounded operator in $B^{\#}$. If |Ψ〉 and |Φ〉 are not in the same sector when N→∞, for any ε>0, one can find a finite set J⊂[1,…,N] of *M* indices αs, all distinct, so as to build $|\Psi_{M}\rangle$ and $|\Phi_{M}\rangle$ as above, and the restriction ${\hat{A}}_{M}$ of $\hat{A}$ to ${⊗}_{\alpha \in J}H_{\alpha }$, such that $|\langle \Psi_{M}|{\hat{A}}_{M}|\Phi_{M}\rangle |<\varepsilon$.

This gradual onset of the properties means that the non-separable, broken-down limit is reached in a controlled manner, at least in the weak topology that is relevant for von Neumann ($W^{*}$-)algebras (note that we make no claim here about $C^{*}$-algebras that rely on a different topology to control the limits because, as described above, our quantum formalism needs projectors and traces, the latter not being necessarily available in $C^{*}$-algebras). This controlled approach to the limit is very much reminiscent of the controlled approach to the Central Limit Theorem at the thermodynamic limit, on which much of equilibrium statistical mechanics relies. Another example is the pervasive function derivative, which considers infinitesimal elements, even though we might think that there is also an ultraviolet cutoff at the Planck scale that makes them no more valid than the thermodynamic limit. On top of this, this gradual onset can be understood in a ’for all practical purposes’ way, in the sense that for a large enough system, the inter-sector coherences in the density operator are so weak that it would take experimental repetitions over more than the age of the Universe to observe a quantum effect in such a system.

*From an epistemological point of view*, this limit can be validated too. As a matter of fact, however generic a conceptual representation of reality may be, it remains a *model* of reality [34]. Generally, physicists assume that:There is a mapping between concepts in the representation (that can be expressed in mathematical language) and the target elements of reality.This mapping allows conducting surrogate reasoning [35] on the concepts to yield (falsifiable) claims on the elements of reality they are meant to describe.

Although this might be blurred in the daily exercise of physics, representations and reality are elements of two different worlds; thus, the conceptual elements of a model are not elements of reality. Moreover, models are, by definition, approximate and valid until proven wrong through an experiment. So, models do not need to have all the properties of reality to be relevant, especially in the most remote corners of their application domain (however, the history of science has shown that unexpected properties of representations could actually have surprising counterparts in reality).

Overall, we consider that these two arguments in two distinct domains validate the consequences that can be derived from taking the N→∞ limit in QM.

### 3.2. Unitarity Relevance and Multiverse Interpretation

We return here to the standard role of unitary transformations in QM, that is, time evolution. Assuming a Hamiltonian $\hat{H}\left(t\right)$ that describes the energy of an otherwise isolated system, it is well known that the system’s evolution can be described by a unitary operator $\hat{U}\left(t\right)$, the solution of the equation $i\hbar \;d\hat{U}\left(t\right)/dt=\hat{H}\left(t\right)\hat{U}\left(t\right)$. More generally, when a transformation (rotation, translation, etc.) is applied at the classical level, one can define a corresponding unitary transformation to be applied to the states or observables of the system. In this point of view, time evolution is just a translation in time, and the Hamiltonian is the infinitesimal generator of such translations. This important subject can be developed in great detail, and it shows the importance of Uhlhorn’s theorem of building representations of symmetry groups [36].

This role of unitarity in time evolution, contrasted with the abrupt evolution during a measurement, is a disturbing situation, especially considering the tremendous success of QM. This becomes even more unsettling when considerations of measurements are extended to the whole Universe, potentially leading to the ’many worlds’ conclusion that there is an infinity of parallel different universes, where each possible outcome of any measurement is realised. In our approach described above, these extrapolations are unwarranted and result from a misunderstanding and misuse of quantum formalism.

Nevertheless, in defence of the idea of multiple parallel universes, it may be said that science has already made many counter-intuitive predictions, establishing, e.g., that the Earth is round and moving quite fast, despite the ’obvious empirical evidence’ that it is flat and motionless. Actually, this view raises two issues of different natures:For it to be a scientific statement, it would need to yield a falsifiable experimental prediction, much like Bell’s inequalities for local hidden variables. Such a prediction is not yet available. Actually, this idea only arises as a consequence of extrapolating the type-I quantum formalism by carelessly applying it to macroscopic systems and then to the whole Universe. This is the difference between the round and moving aspects of the Earth, which quickly led to many practical predictions, e.g., sailing around it, that have largely been vindicated.The above considerations regarding ITP show (if the model holds) that there is no reason to expect any unitarity whatsoever at a macroscopic scale, and thus the very motivation for parallel universes collapses.

But all this is not a real surprise, as the previous section explained that TBQM can be derived from a set of postulates that include the unicity of the Universe.

### 3.3. Reductionism vs. Dualism

Despite all classical objects being built from quantum objects, the radical difference between classical and quantum behaviours raised the question of their compatibility very early in the history of quantum physics. Two antagonistic positions have emerged. Bohr and followers have claimed that there are two fundamentally different levels of reality, separated by a ’Heisenberg cut’, and that physicists have to live with this dual description of reality. Let us call this position ’dualist’. The other position aims to look for a mechanism through which classical behaviours could emerge from quantum ones. One could consider Ehrenfest’s theorem or intense coherent states [11] as the first indications in this direction, which were then further developed in the frame of decoherence theory [12,13]. Classical behaviour would reduce to a part of quantum behaviour in the ’reductionist’ perspective.

The closing of the CSM loop through infinite tensor products sheds new light on this antagonism. In the language of CSM, the dualist approach translates into considering that there is always a (classical) context surrounding a quantum system, with a Heisenberg cut separating them. This assumption is the prerequisite of the contextual quantisation postulate discussed in Section 2.2. These postulates allow for the derivation of the usual TBQM, so duality and contextuality lead to QM. But, forgetting that this usual QM formalism can result from dualism and just taking it for granted, one can see that the key properties of contextuality (the Kochen–Specker theorem) and the difference between quantum and classical behaviours (through von Neumann’s breakdown theorem on ITP) result from the usual QM. So, the algebraically extended QM, as discussed above, implies duality and contextuality. In other words, dualism implies reductionism and reductionism implies dualism. In logical terms, this means that both positions can be viewed as *equivalent* and not antagonistic.

## 4. Conclusions: QM for Engineers and Beyond?

There are currently many engineers working on quantum technologies, and in engineering, it is clear that a reliable physical ontology is extremely useful for determining which objects and properties they are working with. In such a framework, speaking about inaccessible multiple worlds or dead-and-alive cats is not very enlightening; invoking only abstract equations is not much better.

So, returning to our unique world, for the worst and for the best, may be quite useful in practice. Also, rather than stating that a quantum superposition is ‘being in two states at the same time’, it is better to state that one can obtain a result with certainty for some measurement in some context, and a random result for other measurements in another context determined at measurement time. Perhaps the strangest feature of quantum randomness is that it can be turned into a certainty for well-chosen measurements. But, when viewed from an engineering side, this is a quite manageable idea, likely to orient thinking in a practically usable direction.

From a more foundational perspective, it is clear that the views presented here have a strong Bohrian flavour, although they are quite distinct from Bohr’s ideas; for instance, we never speak about complementarity, which is too vague in our opinion. Also, it can be said that our approach aligns closely with the so-called Copenhagen point of view—certainly closer to it than to any other ’interpretation’ of QM. However, the Copenhagen framework is also loosely defined and does not include topics like non-type-I operator algebras that are essential for our construction. As a matter of fact, considering the use of non-type-I algebras allows for proposing a global model where the classical and the quantum realms have a clearly articulated relationship, clarifying how each one relates to the other.

So, overall, we call for an extension of the QM formalism towards operator algebras, despite the known mathematical difficulties within this area. Perhaps this field progressed too quickly into mathematics, so it should be reconsidered by physicists and reintroduced into their domain. 

## Figures and Tables

**Figure 1 entropy-25-01600-f001:**
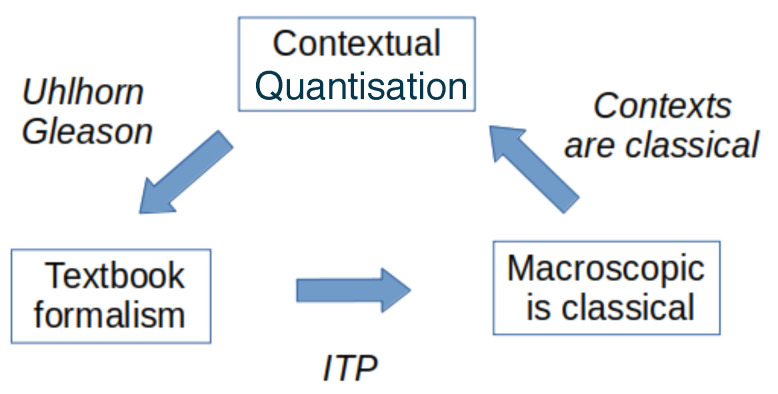
Closing the loop of contexts, systems, and modalities (CSM) using Infinite Tensor Products (ITP).

## Data Availability

Theoretical research, no new data were created.

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
