# Peer review of "Postulating the Unicity of the Macroscopic Physical World"

_entropy, 2023, doi:10.3390/e25121600_

Round 1
Reviewer 1 Report
Comments and Suggestions for Authors
The paper deals with an axiomatization of QM based on Contextuality as developed by one of the authors: after clearly stating and discussing these axioms, the discussion is developed around them in the spirit of obtaining QM with a clear visual interpretation (in the words of the authors, "clear" in the sense of being applicable technologically --- from engineers viewpoint too).
The paper is very well written and makes good and solid arguments for a problem (that of seeing what we are doing in QM) that is still central to this discipline. My only concern, if I really have to find something to comment on, is that there is a number of statements that are told, but only referred to previous works. While this is certainly necessary, too much of this may discourage readers that haven't followed previous material.
But, as I said, it is a matter of taste. I might encourage the authors to add some more explanations for key facts that are better discussed elsewhere, but I also realize that this may make much heavier the reading --- I leave it to their judgment.
Comments on the Quality of English LanguageMinor editing (but very very minor things)
Author Response
We thank the referee for the positive assessment of our work.
We agree that there are many statements that refer to previous works and are not fully detailed. However, there is a difficult balance to find between giving all details - with a risk for the reader to miss the main line of reasoning - and omitting too many details - with the risk of being unclear.
In order to be clear on that, and in agreement with the editor, our article will be categorized as a ‘Perspective’ rather than as a standard article. This allows us to take a broader point of view, with the goal to explain the major issues in our approach, without going into all technical details. We have also reduced the proportion of self-citations, and added more general references. We hope this makes the paper more reader-friendly, and fulfills the referee’s request.
Reviewer 2 Report
Comments and Suggestions for Authors
This paper is very much on the philosophical underpinnings of Grangier’s work on the fundamental ideas and underpinnings of quantum mechanics. There are some issues to address before acceptance.
Please define unicity in the abstract and early on in the text as this word is not commonly used nor is it particularly well-defined.
Line 31. Change don’t to do not.
Please add in more references. For example, see paragraph from lines 49-58. Give references to Bohmian, pilot wave, Everett, etc.. Not everyone will be as well versed in the literature as the authors, especially younger readers. The authors need to go over the entire manuscript and add them in. Please give more references for TBQM than ref 5. Make sure to give references to all theorems, etc.
Line 61. Change macroscopical to macroscopic.
Line 108. Give von Neumann postulate and references.
This is a nice overview of the ideas underpinning Grangier’s work on the fundamentals of quantum mechanics. As it is more of a review, then the authors need to provide better referencing and documentation to the ideas in it so that all readers can delve more deeply into the topic. Not everyone is a well-read on this topic as the two authors.
Comments on the Quality of English Languageacceptable. see above for comments.
Author Response
We thank the referee for the positive assessment of our work and useful comments, that we took into account as follows.
Please define unicity in the abstract and early on in the text as this word is not commonly used nor is it particularly well-defined.
This is an interesting comment, since one may hesitate between ‘unicity’ and ‘uniqueness’, that is used more often. Actually it appears that the meanings are different: according to https://thecontentauthority.com/blog/uniqueness-vs-unicity , unicity refers to the state of being unique in the sense of being the only one of its kind (the relevant meaning here), whereas uniqueness refers to the state of being different from others (for instance, being the best of its kind). We added a reference to explain this.
Line 31. Change don’t to do not.
Done.
Please add in more references. For example, see paragraph from lines 49-58. Give references to Bohmian, pilot wave, Everett, etc.. Not everyone will be as well versed in the literature as the authors, especially younger readers. The authors need to go over the entire manuscript and add them in. Please give more references for TBQM than ref 5. Make sure to give references to all theorems, etc.
There is a challenge there, because there are so many articles on Bohmian, pilot wave, Everett, etc that is is difficult to decide which one to quote. In our opinion Chapter 11 of ref. [2] provides an excellent overview of all interpretations, in a way that is not biased by a particular point of view. There are also many relevant overview articles available on the Standford Encyclopedia of Philosophy. So we added a reference to this chapter 11, and to https://plato.stanford.edu/search/searcher.py?query=quantum+mechanics. We also included more basic references on TBQM (Landau, Feynman) and checked the references to theorems.
Line 61. Change macroscopical to macroscopic.
Done.
Line 108. Give von Neumann postulate and references.
There are several (equivalent) versions of this postulate, so we cite again three TBQM books.
This is a nice overview of the ideas underpinning Grangier’s work on the fundamentals of quantum mechanics. As it is more of a review, then the authors need to provide better referencing and documentation to the ideas in it so that all readers can delve more deeply into the topic. Not everyone is a well-read on this topic as the two authors.
In order to acknowlege the overview character, and in agreement with the editor, our work will be categorized as a ‘Perspective’ rather than as a standard article. This allows us to take a broader point of view, with the goal to explain the major issues in our approach, without going in all technical details. We have also reduced the proportion of self-citations, and added more general references as explained above. We hope this makes our article both useful and self-consistent.